

# A comprehensive review of circRNA: from purification and identification to disease marker potential

Sheng Xu[1], LuYu Zhou[1], Murugavel Ponnusamy[1], LiXia Zhang[2], YanHan Dong[1], YanHui Zhang[1], Qi Wang[1], Jing Liu[1] and Kun Wang[1]

[1] Center for Developmental Cardiology, Institute of Translational Medicine and School of Basic Medicine, Qingdao University, Qingdao, Shandong, China
[2] Department of Inspection, The Medical Faculty of Qingdao University, Qingdao, Shandong, China

## ABSTRACT

Circular RNA (circRNA) is an endogenous noncoding RNA with a covalently closed cyclic structure. Based on their components, circRNAs are divided into exonic circRNAs, intronic circRNAs, and exon-intron circRNAs. CircRNAs have well-conserved sequences and often have high stability due to their resistance to exonucleases. Depending on their sequence, circRNAs are involved in different biological functions, including microRNA sponge activity, modulation of alternative splicing or transcription, interaction with RNA-binding proteins, and rolling translation, and are a derivative of pseudogenes. CircRNAs are involved in the development of a variety of pathological conditions, such as cardiovascular diseases, diabetes, neurological diseases, and cancer. Emerging evidence has shown that circRNAs are likely to be new potential clinical diagnostic markers or treatments for many diseases. Here we describe circRNA research methods and biological functions, and discuss the potential relationship between circRNAs and disease progression.

## INTRODUCTION

Circular RNA (circRNA) was considered as a class of endogenous noncoding RNA (ncRNA) (*Wilusz & Sharp, 2013*), but it is now considered that circRNA can be translated into functional polypeptides (*Legnini et al., 2017*; *Pamudurti et al., 2017*; *Yang et al., 2017b, 2018b*). Unlike linear ncRNA, circRNA is formed with different combinations of sequences and domains, and can be divided into three categories; namely, exonic circRNA (ecRNA) (*Zhang et al., 2014*), circular intronic (ciRNA) (*Zhang et al., 2013*) and exon-intron circRNA (ElciRNA) (*Li et al., 2017b*) (Table 1; Fig. 1). Similar to other ncRNAs, the sequence and structure of circRNA determine its biological functions. CircRNA is mainly located in the cytoplasm and is highly stable compared to other ncRNAs (*Danan et al., 2012*). In addition, recent research has shown that the lengths of mature circRNA dictate the mode of nuclear export (*Huang et al., 2018*). CircRNA is abundantly expressed and evolutionarily conserved across eukaryotic organisms (*Morris & Mattick, 2014*;

Corresponding author
Kun Wang, wangk696@qdu.edu.cn

**Table 1 The characteristics of different types of CircRNA.**

| Name | Type | Location | Joint site | Sequence feature | Function |
|---|---|---|---|---|---|
| ecRNA (*Zhang et al., 2014*) | exon | cytoplasms | 3′–5′ phosphodiester bond | Formed by cyclization of exons containing the reverse complementary sequence of introns and selective cyclization. | Functioning as miRNA sponges; Interact with RNA-binding proteins (RBPs); Participates in translation. |
| CiRNA (*Zhang et al., 2013*) | intron | nucleus | 2′-5′ phosphodiester bond | 5′ splice site enriched with 7 GU motifs and 3′ branch site contains 11 C motifs. | Regulation of gene transcription. |
| ElciRNA (*Li et al., 2017b*) | exon–intron | nucleus | 3′-5′ phosphodiester bond | Formed by cyclization of exons containing the reverse complementary sequence of introns and selective cyclization. | Regulation of gene transcription. |

**Notes:**
ecRNA, exon circRNA; ciRNA, intron circRNA; ElciRNA, exon-intron circRNA.

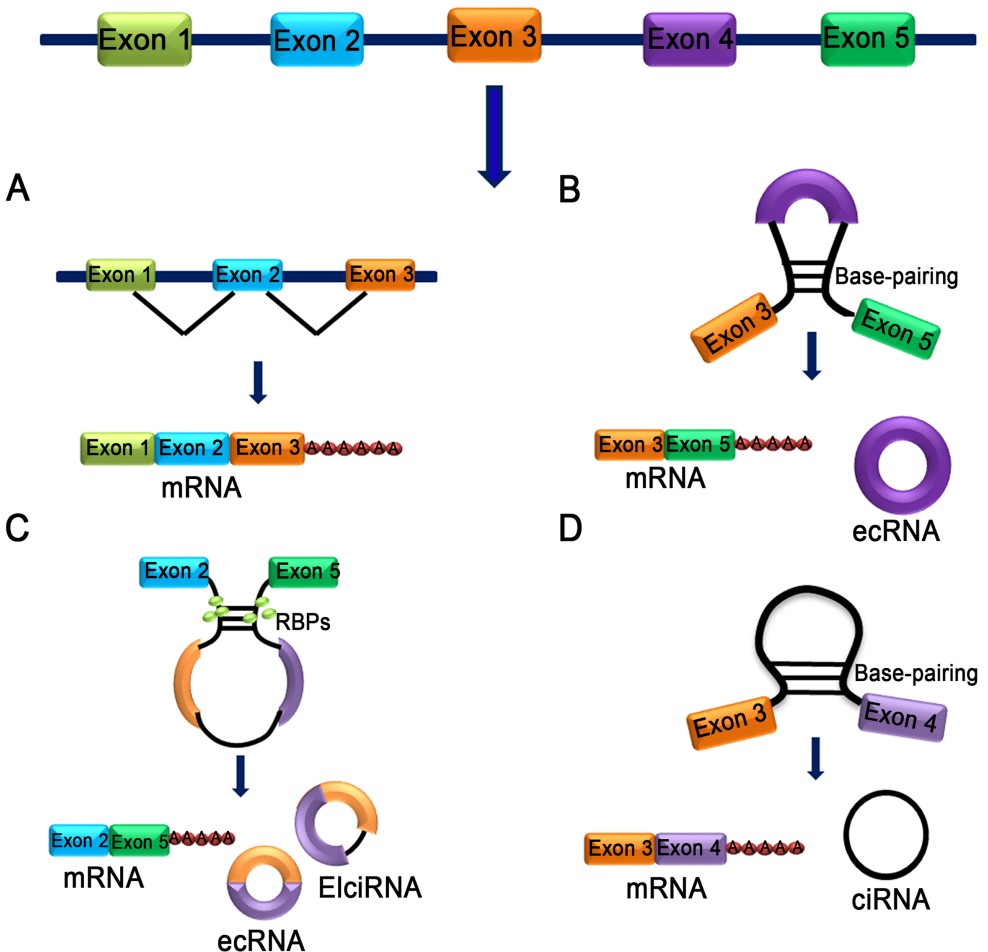

**Figure 1 Characteristics of different types of circRNA.** (A) mRNA: A class of single-stranded ribonucleic acids with genetic information transcribed from deoxyribonucleic acid (DNA). (B) Exon skipping event results in covalently splices and forms an ecRNA after the introns were removed. (C) The interaction between two RBPs can bridge two flanking introns together and form ecRNA, ElciRNA and mRNA. (D) RNA polymerase cleaves the intron from pre-mRNA to form an annulus, the circRNA formed in this manner is ciRNA.
*Wang et al., 2014*), and it plays critical roles in many diseases, including nervous system disorders, cardiovascular diseases (CVDs), diabetes, and cancer (*Burd et al., 2010*; *Wang et al., 2017b*). CircRNA governs gene expression through guiding a number of other molecules, such as splicing factors, RNA polymerase II (*Jeck et al., 2013*), small nuclear ribose nucleoprotein (snRNP) (*Huang & Shan, 2015*; *Li et al., 2017b*) and miRNAs (*Li et al., 2015a*). These interactions promote or inhibit the transcription of the corresponding mRNA.

## SURVEY METHODOLOGY

Analysis: Through extensive literature searches, the role of circRNA in diseases and the methods of circRNA detection and characterization were analyzed, indicating the importance of circRNA and its research prospects, including our previous research results combined with other research results.

## BIOGENESIS OF CircRNA

CircRNA does not have terminal structures, such as a 5′ end cap and 3′ end poly (A) tail, which are covalently closed to form a circular structure (*Jeck et al., 2013*). Jeck and colleagues (2013) proposed two different models of exon circularization. One model is intron-pairing-driven circularization (Fig. 2A), and the other model is lariat-driven circularization (Fig. 2B) (*Jeck et al., 2013*). In the first mechanism, the two introns that flank the exon or exons of the incipient circRNA have a complementary structure to bind to each other. The pairing of the flanking introns brings the splice sites close to each other, shaping a secondary structure that makes back-splicing possible. In the second mechanism, a pre-mRNA is spliced, and two transcripts are produced as follows: an mRNA from which one or more exons are missing; and a lariat consisting of the skipped exons, which makes the circularization possible. The exon lariat is spliced one more time to generate two other elements, namely a circRNA and an intron lariat. Typically, many lasso structures are formed by introns, but they are degraded by the branching enzyme (*Rodriguez-Trelles, Tarrio & Ayala, 2006*).

In these two typical models of circularization, ALU complementary flanking elements (retrotransposons characterized by the action of the *Arthrobacter luteus* (Alu) restriction endonuclease) repeated in intronic regions compete with canonical linear-RNA splicing and act as an inevitable accelerator in the formation of circRNA by reverse complementary matches (*Ashwal-Fluss et al., 2014*; *Hansen et al., 2013*; *Ivanov et al., 2015*; *Jeck et al., 2013*). DExH-Box Helicase 9 (DHX9) is an RNA helicase that specifically binds to reverse Alu elements (IRAlu) to guide the formation of circRNA (*Aktas et al., 2017*). IRAlu has already become a significant basis of analyzing and forecasting the formation mechanism of circRNA (*Zhang et al., 2014*). In addition, the genomic structure of long exons flanked by long introns harboring inverted repeat elements facilitates RNA circularization (*Jeck et al., 2013*). Many proteins are involved in circRNA biogenesis. In normal growing cells, NF90/NF110 binds to A/U-rich elements (including base paired Alu elements) in the introns flanking many exons that yield circRNAs, promoting back-splicing events (*Li et al., 2017a*).

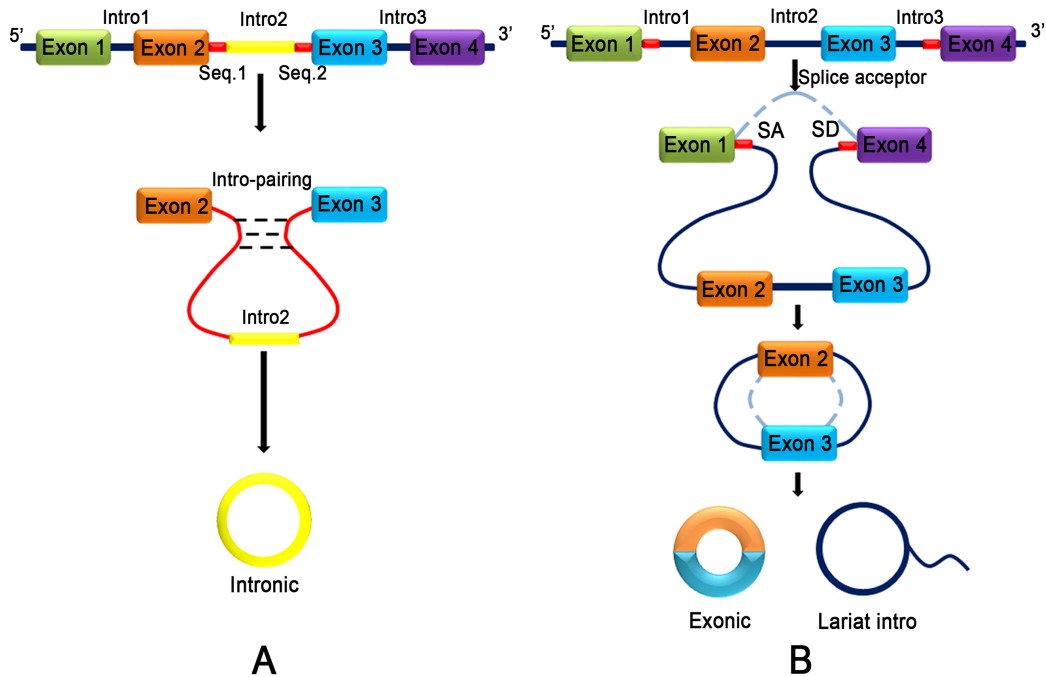

**Figure 2 Two different models of exon circularization of circRNA.** (A) Intron-pairing-driven circularization: during the formation of circRNA, an intron reverse complementary motif comprising GU-rich and C-rich elements is the key component to facilitate cyclization. (B) Lariat-driven circularization: the formation of circRNA is facilitated by the lariat structure. The complementary ALU flanking element which is repeated in the intron region competing for classical linear RNA splicing and the circularization is accelerated by reverse complementarity.

HNRNPL promotes circRNA formation via back splicing (*Fei et al., 2017*). The RNA-binding proteins, such as MBL (muscleblind) (*Ashwal-Fluss et al., 2014*) and QKI (RNA-binding protein quaking I) (*Conn et al., 2015*), also participate in the back-splicing process and cyclization of RNA. Interestingly, high levels of MBL bind to its own pre-mRNA and determine its back-splicing, leading to the inhibition of canonical splicing, decreasing MBL levels and upgrading circMBL (*Ashwal-Fluss et al., 2014*). Monomeric QKI binds to both ends of intron flanking sites and combines to form cyclic exons by bringing the two cyclic shear sites close (*Conn et al., 2015*). FUS regulates circRNA biogenesis by binding the introns flanking the back-splicing junctions (*Errichelli et al., 2017*). CircRNA production is further controlled by FUS (*Errichelli et al., 2017*) and by multiple heterogeneous nuclear ribonucleoprotein (hnRNP) and serine-arginine (SR) proteins (*Fei et al., 2017*; *Kramer et al., 2015*; *Liang et al., 2017*). In contrast, the RNA-editing enzyme, ADARs (Adenosine deaminases acting on RNA) block circRNA formation by binding to complementary double-stranded areas of flanking introns and abolishing the interaction of double-stranded chains (*Ivanov et al., 2015*). Recent research has found that inhibition or slowing of pre-mRNA processing mechanisms, such as spliceosomes, leads to profound increases in circRNA production by extending read through to downstream genes and production of circRNA (*Liang et al., 2017*).
## PROPERTIES OF CircRNA

CircRNA has several unique features and properties when compared to other linear RNAs and ncRNAs. Most of the unique features are generated from exons, while few others are generated from introns or intron fragments (*Cocquerelle et al., 1993*). Several circRNAs possess microRNA response elements (MREs), which enable them to interact with miRNAs to govern target gene expression (*Hansen et al., 2013*; *Yang et al., 2016*). Many circRNAs are derived from pre-mRNA and regulate their own gene expression predominantly at posttranscriptional levels (*Salzman et al., 2012*). Generally, circRNAs show tissue-specific and/or developmental stage-specific expression patterns similar to those of corresponding linear mRNA targets, and their expression level is >10 times higher than that of the linear mRNA (*Jeck et al., 2013*; *Memczak et al., 2013*). CircRNA exists and has been detected in many types of extracellular body fluids, such as saliva, blood and urine (*Jeck et al., 2013*; *Qu et al., 2015*). More than 400 circRNAs have been found in human cell-free saliva from healthy individuals (*Bahn et al., 2015*). CircRNAs have evolutionary conserved sequence features across different species (*Rybak-Wolf et al., 2015*). The covalently closed loop structures lacking 5′–3′ polarity and without poly-adenylated tail favor resistance to RNA exonuclease degradation (*Suzuki & Tsukahara, 2014*). CircRNA plays stable biological roles because the average half-life of circRNA in most species is much longer than its linear counterpart (*Bahn et al., 2015*; *Memczak et al., 2013*).

## FUNCTION OF CircRNA

CircRNA has a variety of functions, including modulation of alternative splicing or transcription, regulating the expression of parental genes, interacting with RNA-binding proteins (RBPs), altering RBP activity, miRNA sponge activity, rolling circle translation and generating pseudogenes.

### CircRNA modulates alternative splicing or transcription

CircRNA participates in the regulation of alternative splicing and transcription, thereby controlling gene expression (Fig. 3A). For example, circMbl is generated from the second exon of the splicing factor MBL, which competes with canonical pre-mRNA splicing, while circMbl and its flanking introns have conserved MBL-binding sites to allow binding to MBL. Interestingly, the alteration of MBL level significantly affects circMbl formation, and this effect depends on MBL-binding sites in the flanking intronic sequences (*Ashwal-Fluss et al., 2014*). Studies have shown that several circRNAs are abundantly found in the nucleus where they regulate transcriptional activity by interacting with polymerase II and homeopathic reactions. For instance, EIciRNA interacts with snRNPs to regulate the transcription of parental genes in a homeopathic manner (*Chen, 2016*). *Li et al. (2015a)* found that circRNA-ITCH interacts with miR-7, miR-17, and miR-214 as well as upregulates the expression of ITCH. During embryogenesis, sisR-4 promotes transcription of its host gene by activating an enhancer present in the intron where sisR-4 is encoded, which is essential for development (*Tay & Pek, 2017*). HNRNPL directly regulates the alternative splicing of RNAs, including encoding the androgen receptor, the key lineage-specific prostate cancer oncogene (*Fei et al., 2017*).

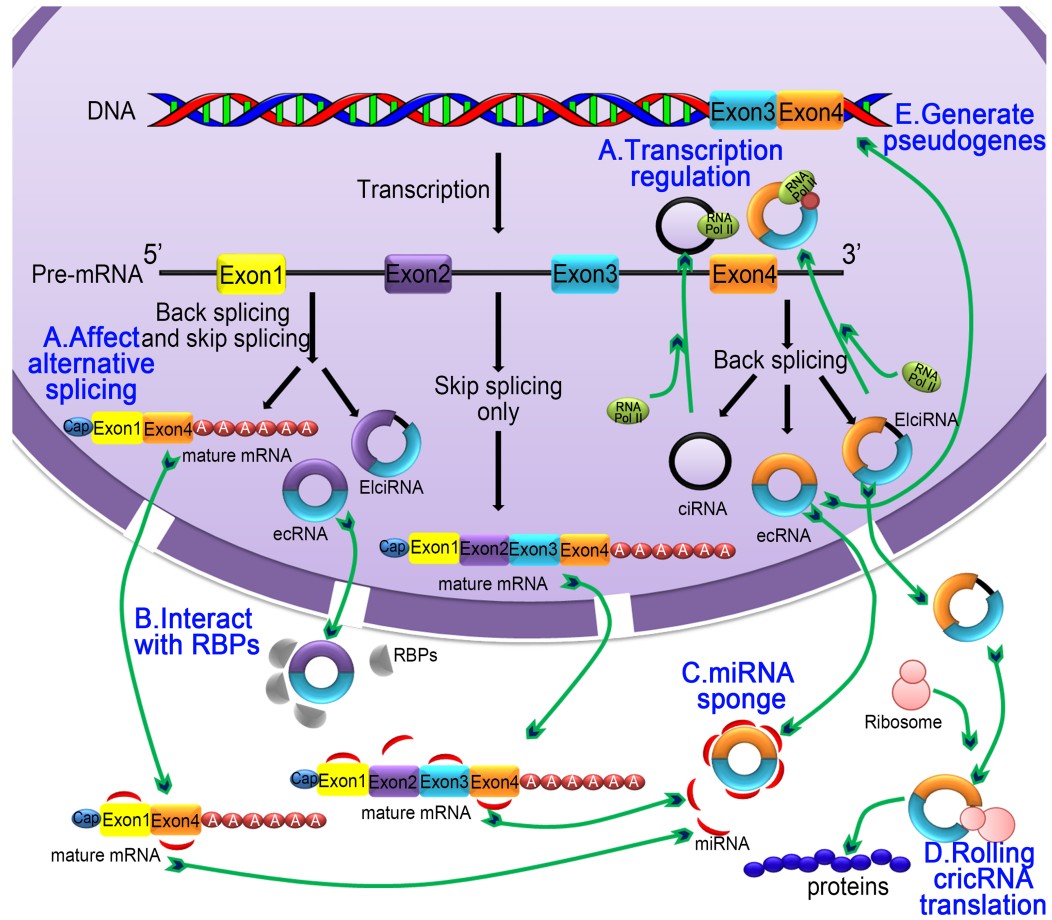

**Figure 3 The five main functions of the circRNA.** (A) Regulating selective splicing or transcription: Stable circRNA and EIciRNAs are located in the nucleus, where they bind to RNA polymerase and promoting transcription; circRNA competes with pre-mRNA splicing to reduce the level of linear mRNA and excludes specificity from pre-mRNA by changing the composition of processed mRNA; (B) interaction with RBPs: circRNA binds with RBPs and ribonucleoprotein complexes and interfere with their functions. As a single circRNA can bind with multiple units of RBPs, they serve as stores of RBPs; (C) miRNA sponging activity: circRNA binds with miRNA and affecting the miRNA-dependent target gene suppression; (D) rolling circle translation: some circRNA can be translated into proteins by means of a roll loop amplification mechanism; (E) generation of pseudogenes: some circRNA are reverse transcribed into cDNA and integrated into the genome; however, the mechanism of integration is not yet clear.            

## CircRNA interacts with RBPs

Apart from miRNA regulation, circRNA can sequester RBPs and thus control the intracellular localization and transport of RBPs and associated mRNAs (*Hentze & Preiss, 2013*; *Jeck & Sharpless, 2014*) (Fig. 3B). Some circRNAs combine with RBPs and ribonucleoprotein complexes, thereby inhibiting their activity. However, circRNA functions as stores of RBPs and ribonucleoprotein complexes. EcRNA acts as a scaffold by specifically binding to protein molecules to provide an interaction platform for RNA-binding proteins, RNA, and DNA. For example, in HEK293 cells, CDR1as contains a region near perfectly complementary to miR-671, which causes the circRNA to be
endonucleolytically cleaved by argonaute 2 (AGO2) in a miR671-dependent manner (*Hansen et al., 2011*). CDR1as is derived from an antisense long ncRNA (*Barrett et al., 2017*) and is expressed several orders of magnitude higher than cerebellar degeneration-related protein 1 (CDR1) gene from the opposite strand (*Hansen et al., 2011*; *Piwecka et al., 2017*). *Chen et al. (2017d)* found that there are cellular differential mechanisms in the recognition of internal and external circRNAs as follows: external circRNA induces activation of RIG-I-mediated cellular autoimmune effector pathways, and endogenous circRNA does not induce this pathway due to binding of RBPs.

## CircRNA as MiRNA sponge

CircRNAs act as competing endogenous RNAs (ceRNAs) that contain shared MREs by which they sequester miRNAs and prevent their interactions with target mRNAs (Fig. 3C). Systematically validated circRNAs, such as ciRS-7 (CDR1as) (*Hansen et al., 2013*; *Memczak et al., 2013*) and Sry circRNA (*Hansen et al., 2013*; *Zhao & Shen, 2015*), are produced from the mRNAs of CDR1 and dysregulated rat testis SRY, respectively. During the embryonic developmental process in zebrafish, the expression of CDRlas reduces brain volume, thereby hampering brain development. However, exogenous delivery of miR-7 reverses the brain volume reduction and reinstates normal brain development, indicating that CDRlas blocks miR-7 by sponging functions (*Hansen et al., 2013*; *Hansen, Kjems & Damgaard, 2013*). New research has found that the CDR1as sequence overlaps the lncRNA LINC00632 sequence (*Barrett et al., 2017*). In general, there are only several circRNAs containing enough miRNA-binding sites to function as strong sponges, and other circRNAs are exceptional cases (*Chen, 2016*; *Tay & Pek, 2017*). Knockdown of circHIPK2 expression significantly inhibits astrocyte activation via regulation of autophagy and endoplasmic reticulum stress through targeting MIR124-2HG and SIGMAR1 (*Huang et al., 2017*). CircHECW2 plays a role in the epithelial-mesenchymal transition (EMT) pathway by competitively inhibiting miR-30D, which releases ATG5, thereby promoting the Notch1 signaling pathway (*Yang et al., 2018a*).

## Rolling circle translation

In eukaryotic cells, cyclic mRNA can be translated by typical translation machineries because it contains an internal ribosome entry site sequence, and it can bind directly to the ribosome (*Thompson, 2012*) (Fig. 3D). In prokaryotic cells, such as *E. coli*, circRNA contains a well-conserved infinite open reading frame (ORF) system, which enables the translation of circularized RNA (*Abe et al., 2015*). In eukaryotic systems, some circRNAs have binding sites for ribosomal 40S subunits, thus initiating translation, which has been demonstrated both in vivo and in vitro studies (*Holdt, Kohlmaier & Teupser, 2017*). In an *E. coli* system, circRNA with green fluorescent protein (GFP) inserted in the ORF can successfully translate GFP (*Wang & Wang, 2015*). Interestingly, circRNA also drives protein translation by methylation of adenosine N6 (m6A) (*Yang et al., 2017b*). Protein translated by circRNA can act synergistically with the protein expression products of the parent gene and function together. For example, circ-FBXW7

translates a new protein that inhibits glioma (*Yang et al., 2018b*). Circ-ZNF609 directly translates into proteins that participate in muscle formation (*Legnini et al., 2017*). In prokaryotic cells, proteins are generated from circRNA by means of rolling circle amplification analogous to a polymerase reaction in the eukaryotic translation system, which reveals that there is no need for multiple binding of translational machinery to the RNA template (*Rodriguez-Trelles, Tarrio & Ayala, 2006*). The circular amplification not only produces long and repetitive peptide sequences but also increases the productivity of the linear counterpart (*Thompson, 2012*).

### Generate pseudogenes

Studies have shown that stable circulatory molecules can be reverse transcribed and integrated into the genome to form circRNA-derived pseudogenes (*Dong et al., 2016*) (Fig. 3E). Bioinformatics analysis of the mouse genome using computational pipeline (CIRCpseudo) found that at least 33 pseudogenes are possibly derived from the same circRNA at the ring finger and WD repeat domain 2 (RFWD2) locus (circRFWD2) and that nine of the pseudogenes are from exons (exons 2 to 4 or 5) of circRFWD2. It is well documented that pseudogenes play an important role in cell differentiation and in cancer progression (*Kalyana-Sundaram et al., 2012*).

## METHODS OF CircRNA DETECTION AND CHARACTERIZATION

### Preliminary purification and identification

#### Molecular biology method

The loop structure of circRNA has high stability compared to linear RNA, and it is resistance to enzyme digestion (*You & Conrad, 2016*). Therefore, an enzymatic digestion method can be used for the preliminary purification and identification of circRNA (*Jeck & Sharpless, 2014*).

First, the processing of extracted RNA with exonucleases, such as RNase R, nicotinic acid phosphatase and 5′ end exonuclease, destroys most linear RNA, but circRNA remains intact due to no open ends in circRNA for these enzyme reactions. A circRNA-specific divergent primer can be used to amplify abundant circRNA in which linear RNAs do not amplify (*Jeck & Sharpless, 2014*; *Suzuki et al., 2006*). Second, the migration velocity of circRNA is slower than that of long linear RNA due to lack of polarity at the end. Particularly, circRNA migration is much slower than RNA from homologous gene transcription in weak crosslinked gels, and this difference helps to detect circRNAs easily through Northern blot analysis (*Tabak et al., 1988*). Third, the fluorescence in situ hybridization technique can locate circRNA at a subcellular level (*Li et al., 2017b*; *Zhang et al., 2013*). As circRNA does not have a poly (A) structure, the traditional oligo dT enrichment method using a Ribo-Zero kit to remove rRNA is not effective. The removal of linear RNA using RNase R is the most effective step for the enrichment of circRNA and generating a circRNA library (*Ebbesen, Kjems & Hansen, 2016*; *Jeck & Sharpless, 2014*).

### High-throughput sequencing

The traditional RNA-Seq technique does not distinguish circRNAs from linear RNAs (*You & Conrad, 2016*). As a result, improvements have been made to detect and validate circRNA. First, as the intergenic exon rearrangement has different forms, generation of divergent primers with boundary combinations can form circRNA candidate sequence, which can be used to compare to sequencing data (*Salzman et al., 2013*). Second, bioinformatics analysis of whole genome sequence and assessment of sequence data through different sequence alignment algorithm can be used to identify circRNAs (*Jeck et al., 2013*). Third, templates designed with multiple sequence splice joints can directly detect circRNAs from cDNA sequence (*Hoffmann et al., 2014*). Currently, many algorithms are available for the prediction and study of circRNAs, including Acfs (*You & Conrad, 2016*), FUCHS (*Metge et al., 2017*) and CIRI2 (*Gao, Zhang & Zhao, 2017*). Acfs allows accurate and fast identification of circRNA, and it also determines the abundance of circRNAs from single- and paired-ended RNA-Seq data. Acfs is well suitable for a wide spectrum of applications, including characterizing the landscape of circRNA from a variety of organisms. The FUCHS system is based on long sequencing reads (>150 bp/reads), which detects circRNA within the variable shear and provides other information for more accurate interpretation. CIRI2 uses the maximum likelihood estimate based on multiple seed matches to identify reverse splice junctions, and it filters out false positives and mapping errors derived from the repetitive sequence. CIRI2 has a significant balance of sensitivity, reliability, duration, and RAM usage (*Ebbesen, Kjems & Hansen, 2016*; *Jeck & Sharpless, 2014*).

### Gene chip

The human genome array, U133plus2.0 tool, can detect mRNA but cannot detect ring RNA because the probe is designed for linear RNA (*Lu et al., 2017*). Therefore, the human genome array cannot effectively distinguish circRNA and linear RNA when a normal probe is used. However, if the probe is designed based on the reverse splice site of the circRNA, this array tool can specifically detect circRNA because there is no reverse splice site sequence in linear RNA, thus effectively distinguishing circRNA and mRNA (*Li et al., 2018*).

### Primer design

Recently, the field of circRNA research is gaining more attention because circRNAs contribute to many physiological and pathological processes. Unlike conventional PCR primers, the design of circRNA primers should consider certain criteria. For the detection of ecRNA, primers should be designed for the cross-cut site (back-splice). In the case of ciRNA, primers targeting cross-cleavable sites should be used. Primers can also be designed around intron regions. Moreover, the length of the amplified product should not be more than 100 bp. Sequence position transformation is also important (*Panda & Gorospe, 2018*). The differences in the selection of primers for linear RNA and circRNA are listed in Fig. 4.

The actual amplification effect after primer design needs to be experimentally determined. If the quantification of circRNA is performed by qPCR, the length of amplification should be

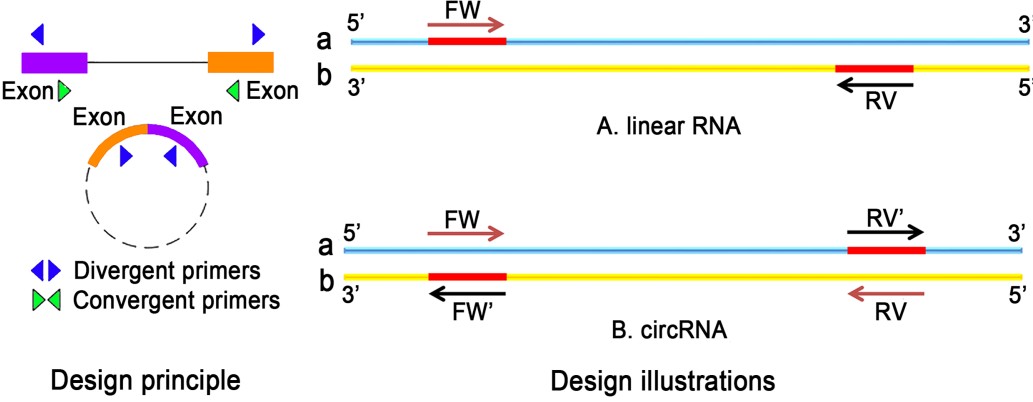

**Figure 4 The difference between linear RNA and circRNA primer design.** (A) FW is a forward primer with the b chain as template. The base sequence of synthesis is the original sequence of a. RV is a reverse primer with a chain as template, and the base sequence of synthesis is the original sequence of b. The sequence between FW and RV is high; (B) The original primers need to reverse: the synthetic primers are FW′ and RV′, where FW′ is the reverse complementary sequence of the RV primer, RV′ is the reverse complementary sequence of FW primer.                             

identified according to the experimental requirements of qPCR. Thus, qPCR remains the most widely used technique to assess the expression level of circRNA.

## CircRNA research database

In recent years, increasing numbers of circRNA research tools with different aspects and improved functional analysis have been generated. The currently available online databases for the detection and characterization of circRNA, which contain GenBank annotations or circRNA from published articles, are presented here. Each database analyzes circRNA with different characteristic features for detection, and each database provides abundant information for circRNAs. There are several free online databases available for circRNA research as shown in Table 2.

## CircRNA AS A DISEASE MARKER IN DISEASE PROGRESSION

The best-known circRNA, CDR1as, is the inhibitor of miR-7, and it is the critical ncRNA known to be involved in various diseases, including cancer, neurodegenerative diseases, diabetes, and atherosclerosis (*Peng, Yuan & Li, 2015*). In addition more functions of circRNAs being revealed, the underlying relationships between circRNAs and various diseases have been rapidly elucidated. The great specificity and conservation of circRNAs in various tissues add a further dimension to the discovery of these disease biomarkers. CircRNAs involved in diseases are listed in Table 3.

## CircRNA in cardiovascular diseases

Cardiovascular diseases poses an increasing threat to human health. According to a report from the World Health Organization (WHO), nearly 17.5 million people die of CVD each year (*Mendis, Davis & Norrving, 2015*). CircRNA is highly specific and is exclusively expressed in different tissues, including vascular and heart tissue (*Fan et al., 2017*).

**Table 2 Database for circRNA research.**

| Tool name | The latest version | URL | Remarks |
|---|---|---|---|
| circlncRNAnet (*Wu et al., 2018b*) | May 2017 | http://app.cgu.edu.tw/circlnc/ | It aims to broaden the understanding of ncRNA candidates by testing in silico several hypotheses of ncRNA-based functions on the basis of large-scale RNA-seq data. |
| starBase v2. 0 (*Li et al., 2014b*) | December 2013 | http://starbase.sysu.edu.cn | Includes microRNA, mRNA, lncRNA and other RNA information. It is a useful tool for detecting miRNA-circRNA interaction. If there is a need to retrieve all circRNA in the genome, circRNABase is useful. |
| circBase (*Glazar, Papavasileiou & Rajewsky, 2014*) | December 2015 | http://www.circbase.org/ | Thousands of circRNAs are annotated from eukaryotic cells. |
| circ2Traits (*Hancock, 2015*) | December 2013 | http://gyanxet-beta.com/circdb | Provides more information about the genomic positions of circRNAs and circRNA-associated diseases. |
| nc2Cancer (*Cheng et al., 2015*) | | http://bioinfo.au.tsinghua.edu.cn | |
| DeepBase v2. 0 (*Zheng et al., 2016*) | November 2015 | http://rna.sysu.edu.cn/deepBase/ | This database is a platform for annotation and discovery of small (microRNA, siRNA and piRNA) and long ncRNAs from next-generation sequencing data. |
| CircInteractome (*Dudekula et al., 2016*) | December 2015 | https://circinteractome.nia.nih.gov/index.html | This database can be used only to match the circRNA with relevant RNA-binding proteins. |
| TSCD(*Xia et al., 2016a*) | August 2016 | http://gb.whu.edu.cn/TSCD/ | It is useful for characterizing tissue-specific circRNAs in human and mouse genomes. |
| CIRCpedia (*Zhang et al., 2016*) | January 2015 | http://www.picb.ac.cn/rnomics/circpedia/ | This database contains reverse splicing and variable splicing sites of circRNAs from 39 individuals and mouse samples. |
| circRNADb (*Chen et al., 2016*) | | http://reprod.njmu.edu.cn/circrnadb | It contains a record of more than 30,000 exons with circRNA nature in the human genome |

**Note:**
Every database present has its own sphere of competence, and only the perfect combination of various databases can provide accurate information.

### Pathological hypertrophy and heart failure

MiR-223 is an endogenous regulator of hypertrophy in cardiomyocytes, which can induce cardiac hypertrophy and heart failure (HF) (*Wang et al., 2015b*). In cardiac hypertrophy, ARC (apoptosis repressor with CARD domain) is a miR-223 downstream target (*Wang et al., 2016*). The heart-related circRNA (HRCR) can function as an endogenous miR-223 sponge to inhibit miR-223 activity, subsequently increasing the expression of ARC (*Wang et al., 2016*). However, the expression of HRCR is decreased during cardiac hypertrophy and HF. Thus, it is speculated that increased expression of HRCR attenuates the development of cardiac hypertrophy and HF, and it may be an attractive therapeutic target for cardiovascular disorders associated with pathological hypertrophy (*Wang et al., 2016*).

### Atherosclerosis

Circular ANRIL RNA (circular antisense ncRNA in the INK4 locus, cANRIL) is an antisense transcript from the INK4A/ARF (cyclin-dependent kinase 4 inhibitor, INK4a;

**Table 3** CircRNA in disease development and progression.

| Diseases | | CircRNA | References |
|---|---|---|---|
| CVD | Pathological hypertrophy and HF | HRCR | *Wang et al. (2016), Wang & Wang (2015), Wang et al. (2015b)* |
| | Atherosclerosis | CANRIL | *Holdt et al. (2016)* |
| | Cardiac senescence | Circ-Foxo3 | *Du et al. (2016a, 2016b)* |
| | MI | CDRlas | *Geng et al. (2016), Li et al. (2014a), Read et al. (2014)* |
| | | MFACR | *Wang et al. (2017a)* |
| Neurodegenerative diseases | | CDRlas | *Zhao et al. (2016)* |
| | | CircRar1 | *Nan et al. (2017)* |
| Diabetes | | CDRlas | *Kim et al. (2017), Wang et al. (2013), Xu et al. (2015)* |
| OA | | Hsa_circ_0005105 | *Wu et al. (2017)* |
| MDD | | Hsa_circRNA_103636 | *Cui et al. (2016)* |
| Silicosis | | CircHECTD1 | *Zhou et al. (2018)* |
| Cancer | GC | Hsa-circ-002059 | *Li et al. (2015c)* |
| | | Circrna_100269 | *Zhang et al. (2017)* |
| | | Hsa_circ_0003159 | *Tian et al. (2017)* |
| | | Hsa_circ_0000190 | *Chen et al. (2017c)* |
| | | CircPVT1 | *Chen et al. (2017a)* |
| | CRC | Hsa_circ_001988 | *Wang et al. (2015a)* |
| | | Circ_001569 | *Xie et al. (2016)* |
| | | CircCCDC66 | *Hsiao et al. (2017)* |
| | | Hsa_circ_000984 | *Xu et al. (2017b)* |
| | ESCC | Has_circ_0067934 | *Xia et al. (2016b)* |
| | | Cir-ITCH | *Li et al. (2015a)* |
| | HCC | CircZKSCAN1 | *Yao et al. (2017)* |
| | | CDRlas | *Xu et al. (2017a)* |
| | | Hsa_circ_0005075 | *Shang et al. (2016)* |
| | | Hsa_circ_0004018 | *Fu et al. (2017)* |
| | | Hsa_circ_0001649 | *Qin et al. (2016)* |
| | | CircARSP91 | *Shi et al. (2017)* |
| | | Circ-10720 | *Meng et al. (2018)* |
| | | Circ-ITCH | *Guo et al. (2017)* |
| | Cervical Cancer | CDRlas | *Lee et al. (2015)* |
| | BC | Circ-Amotl1 | *Yang et al. (2017a)* |
| | Human oral squamous cell carcinomas (OSCC) | Circrna_100290 | *Chen et al. (2017b)* |
| | Lung adenocarcinoma (LAC) | Hsa_circ_0013958 | *Zhu et al. (2017)* |
| | Bladder carcinoma | CircTCF25 | *Zhong, Lv & Chen (2016)* |
| | | CircPTK2 | *Xu et al. (2017c)* |

alternative reading frame, ARF) locus (*Salzman et al., 2013*), which inhibits the expression of INK4/ARF (*Burd et al., 2010*). Another research group found that cANRIL prevents rRNA prebinding and exonuclease-mediated rRNA maturation by binding to the C-terminal lysine-rich domain of PES1, inducing an increase in the expression and activity of p53 and subsequent decrease in apoptosis. This pathway inhibits atherosclerosis by eliminating hyperproliferative cell types in atherosclerotic plaques, indicating that cANRIL may be associated with the prevention or treatment of atherosclerosis (*Holdt et al., 2016*).

### Cardiac senescence

CircFoxo3 is generated from Foxo3, a member of the forkhead family of transcription factors that is highly expressed in aged heart samples from elder patients. The expression of circFoxo3 is highly correlative with markers of cellular senescence (*Du et al., 2016a*). Experimental studies have found that cells expressing high levels of circFoxo3 are unable to transition to S phase, revealing that circFoxo3 represses cell cycle progression and cell proliferation (*Du et al., 2016b*). CircFoxo3 is mainly distributed in the cytoplasm where it interacts with several transcription factors [E2F1, Focal adhesion kinase (FAK), and HIF1a] and antisenescence proteins, such as ID-1, thus preventing their nuclear entry. In fact, the nuclear entry of FAK and HIF1a is essential for their antisenescence role. Thus, circFoxo3 repress their antiaging effects. In addition, circFoxo3 also positively correlates with cellular senescence (*Du et al., 2016a*). Together, these studies suggest that circRNA originating from FOXO genes may be a promising target for repositioning of ID-1, E2F1, FAK, and HIF1a from the cytoplasm to nucleus, ultimately attenuating cellular senescence in aging hearts (*Du et al., 2016a*).

### Myocardial infarction

Myocardial infarction (MI) is a fatal disease worldwide (*Mozaffarian et al., 2015*). Due to the lack of available biomarkers, MI cannot be predicted effectively (*Vausort et al., 2016*). To date, some studies have made significant progress in solving this issue.

*Wang et al. (2017a)* found that mitochondrial fission and apoptosis-related circRNA (MFACR) plays a protective role in the heart through attenuating mitochondrial fission in cardiomyocytes by directly targeting miR-652-3p and increasing the expression of its target, MTP18, which promotes cell survival in cardiomyocytes. Thus, MFACR-dependent inhibition of miR-652-3p increases MTP18 and mitochondrial fission, which results in a reduction in cardiomyocyte apoptosis and extension of MI injury (*Wang et al., 2017a*). Cdr1as play detrimental roles in MI by acting as a miR-7 sponge and inhibiting its activity (*Geng et al., 2016*; *Memczak et al., 2013*; *Xu et al., 2017a*). It is well documented that miR-7a/b plays a protective role through negatively regulating the expression of poly ADP-ribose polymerase (PARP) and decreasing apoptosis in myocardial cells (*Fan et al., 2017*; *Zhao et al., 2009*). SPI and PARP play proapoptotic roles during MI development (*Li et al., 2014a*; *Read et al., 2014*). Hypoxia-induced increases in SP1 and PARP expression cause apoptotic cell death, and SP1 and PARP are target genes of miR-7, which can decrease cell apoptosis (*Geng et al., 2016*; *Li et al., 2014a*; *Read et al., 2014*).

Collectively, the ciRS-7-miR-7-PARP/SP1 axis may offer new biomarkers for the diagnosis of MI, while additional diagnostic markers remain to be discovered (*Fan et al., 2017*).

## CircRNA in neurodegenerative diseases

There are thousands of circRNAs expressed in brain tissue (*Rybak-Wolf et al., 2015*; *You et al., 2015*). As the central nervous system ages, age-related circRNAs accumulate in the brain and have been identified as promising indicators of aging (*Kumar et al., 2017*). This section focuses on aging diseases, such as nerve injury, Alzheimer's disease (AD) and Parkinson's disease.

In hippocampal neural cells (HT22), *Lin et al. (2016)* found that oxygen-glucose deprivation/reoxygenation (OGD/R) injury significantly regulates the expression of 15 circRNAs compared to normal cells, suggesting the involvement of circRNA in nerve injury. Moreover, CDRlas plays a protective role by inhibiting miR-7, which directly regulates the expression of a-synuclein and ubiquitin protein A. Importantly, a-synuclein and ubiquitin protein A are associated with the occurrence of AD (*Lukiw, 2013*) and Parkinson's disease (*Hancock, 2015*). However, the disturbance of the miRNA-circRNA system in the hippocampal CAl region of disseminated AD causes the adsorption of miR-7, leading to increased expression of ubiquitin protein A in the human central nervous system (*Zhao et al., 2016*). In lead-induced neuronal apoptosis, circRar1 directly inhibits miR-671, resulting in suppression of Akt2 and increased expression of caspase-8 and other apoptosis-related proteins (*Nan et al., 2017*). Currently, the function of circRNA in the nervous system is largely unclear, and the potential of circRNA as a biomarker for neurodegenerative disorder requires further study (*Kumar et al., 2017*).

## Diabetes

CDRlas plays an essential role in islet cell function and insulin secretion (*Xu et al., 2015*). Thus, CDRlas may gain importance in the diagnosis and treatment of diabetes mellitus. It is well known that the impairment of islet $\beta$-cell function leads to absolute or relative insulin deficiency (insulin resistance), which increases blood sugar level and diabetes (*Kim et al., 2017*; *Xu et al., 2015*). MiR-7 negatively regulates islet $\beta$-cell proliferation. MiR-7 overexpression damages the dedifferentiation ability of $\beta$ cells, leading to downregulation of insulin secretion and ultimately diabetes. MiR-7 targets multiple components of the mammalian (mTOR) signaling pathway, which are involved in pancreatic $\beta$-cell proliferation. Silencing of miR-7 expression in $\beta$ cells increases their proliferative activity, indicating that miR-7 affects pancreatic $\beta$-cell renewal and is associated with diabetes mellitus. Together, these findings reveal that CDR1as/miR-7 may be a potential therapeutic target for treating and managing diabetes (*Wang et al., 2013*).

## CircRNA in cancer

Accumulating evidence has shown that circRNAs affect the invasive characteristics of tumors in various ways, including competition with miRNAs, translation into proteins, activity as miRNA reservoirs, and formation of fusion circRNAs (f-circRNAs) (*Han et al., 2017*; *Memczak et al., 2013*). Genomic alterations, particularly aberrant

chromosomal translocations, are responsible for the onset of many types of cancers and solid tumors (*Jeck et al., 2013*). F-circRNA, which is produced from transcribed exons of translocated chromosomes, promotes carcinogenesis by increasing cell viability and resistance to therapy. Abnormal f-circRNA is functionally related to cancer progression in many types of malignancies (*Guarnerio et al., 2016*). The specific roles of circRNAs in various tumors are described in the following sections.

### CircRNA in gastric cancer

Gastric cancer (GC) is the fourth most common gastrointestinal malignant neoplasm and is the third leading cause of cancer-related deaths worldwide (*Doi et al., 2015*). Numerous circRNAs are abnormally expressed in GC. Increased expression of Hsa-circ-002059 is significantly associated with the tumor stage of GC (*Li et al., 2015c*). CircRNA_100269 suppresses gastric tumor cell growth by targeting miR-630. However, circRNA_100269 expression is downregulated during GC and can be used as a biomarker to predict cancer recurrence (*Zhang et al., 2017*). The expression levels of hsa_circ_0003159 (*Tian et al., 2017*), hsa_circ_0001895 (*Shao et al., 2017*) and hsa_circ_0000190 (*Chen et al., 2017c*) are down-regulated in GC. Another research group has found that the expression of circPVT1 is often upregulated in GC tissue (*Chen et al., 2017a*). *Sui et al. (2017)* found that the expression of tumor-associated genes, such as CD44, CXXC5, MYH9, and MALAT1, is regulated through different mechanisms of circRNA-miRNA-mRNA interactions. Together, all studies have shown that circRNA plays a crucial role during GC development and that circRNA expression levels can be used as potential biomarkers for clinical prognosis prediction, sensitivity and specificity (*Li et al., 2015b*).

### Colorectal cancer

In colorectal cancer (CRC), the downregulated hsa_circ_001988 is associated with differentiation and perineural invasion (*Wang et al., 2015a*). Perineural invasion is a predictor of prognosis in colorectal cancer, and it is negatively correlated with survival time and local recurrence in colorectal cancer patients (*Peng et al., 2011*). These results suggest that circRNAs may be potential candidates for therapeutics and biomarkers for CRC (*Wang et al., 2015a*). Evidence has shown that circRNA is related to CRC. Cir-ITCH is also downregulated in CRC, exhibiting an anticancer effect by binding to miR-7 and miR-20a (*Huang et al., 2015*). Circ_001569 directly inhibits the regulatory activity of miR-145, thereby upregulating the expression of its targets, such as E2F5, BAG4, and FMNL2, which are involved in tumor proliferation and invasion in CRC (*Xie et al., 2016*). CircCCDC66 regulates a subset of oncogenes, which control multiple pathological processes, including cell proliferation, migration, invasion, and anchorage-independent growth, in CRC (*Hsiao et al., 2017*). In addition, other circRNAs, such as hsa_circ_000984 (*Xu et al., 2017b*) and hsa_circ_001988 (*Wang et al., 2015a*), are also abnormally expressed in CRC.

### Esophageal squamous cell carcinoma

Esophageal squamous cell carcinoma (ESCC) is one of the most prevalent and deadly types of cancers, and the prognosis of ESCC remains poor (*Xia et al., 2016b*). In ESCC,

has_circ_0067934 is upregulated and accelerates malignant cell proliferation (*Xia et al., 2016b*). The expression of Cir-ITCH inhibits ESCC proliferation by suppressing the Wnt/β-catenin pathway through sponging miRNAs, such as miCH-7, miR-17, and miR-214. It is well known that ITCH mediates degradation of activated Dvl2, which is a key component of the Wnt pathway. However, the downregulation of Cir-ITH in ESCC releases the brakes on the Wnt pathway by enhancing the expression of oncogenic miCH-7, miR-17, and miR-214, consequently leading to uncontrolled proliferation of ESCC (*Li et al., 2015a*). Sun et al. constructed a circRNA-miRNA interaction network, in which circRNA9927-NBEAL1 represents the largest node. These findings indicate that some circRNAs may be novel potential biomarkers and therapeutic targets of ESCC (*Xia et al., 2016b*).

### Hepatocellular carcinoma

Hepatocellular carcinoma (HCC) is the second leading cause of cancer-related deaths across the world and is particularly prevalent in less developed countries (*Doi et al., 2015*). Increasing evidence has suggested that circRNAs may play a key role in the development of HCC. The ZKSCAN1 gene and its related circRNA (circZKSCAN1) inhibit HCC cell growth, migration, and invasion by blocking several signaling pathways (*Yao et al., 2017*). MiR-7 is a tumor-suppressing ncRNA, which attenuates HCC proliferation, and it decreases the risk of microvascular invasion by suppressing the expression of its target gene, PIK3CD, and p70S6K (*Xu et al., 2017a*). However, miR-7 activity is counteracted by the overexpression of CDRlas, which adsorbs miR-7 (*Xu et al., 2017a*). Similarly, hsa_circ_0005075 participates in cell adhesion during HCC development and is considered as a biomarker for HCC (*Shang et al., 2016*). In contrast, the expression levels of tumor suppressive circRNA, such as hsa_circ_0004018 (*Fu et al., 2017*), hsa_circ_0001649 (*Qin et al., 2016*) and CircARSP91 (*Shi et al., 2017*), are significantly downregulated in HCC. CircRNA can interact with transcription factors. Twist is a critical EMT-inducing transcription factor that increases expression of Vimentin, and circ-10720 knockdown counteracts the tumor-promoting activity of Twist1 in vitro (*Meng et al., 2018*). More importantly, circ-ITCH not only has prognostic significance but can also be used as a predictive biomarker for HCC (*Guo et al., 2017*).

### Cervical cancer

Cervical cancer is one of the most common death-causing malignancies in women worldwide. FAK, which is a key regulator of growth factor receptor- and integrin-mediated signal pathways, promotes the proliferation, invasion and migration of cervical cancer cells, and it exacerbates the progression of the disease (*Lee et al., 2015*). In HeLa and C33A cells, the increased level of CDRlas promotes FAK expression by inhibiting miR-7, which targets FAK and acts as a tumor suppressor in cervical cancer cells. This finding indicates that there is a relationship between CDRlas and oncogenic transcription factors in cervical cancer. Thus, the targeted therapy of CDRlas regulatory networks would provide a new approach for the diagnosis and treatment of cervical cancer (*Lee et al., 2015*).
### Breast cancer

Breast cancer (BC), the most frequently diagnosed cancer in women around the world, has been the focus of major advances in the last few decades (*Doi et al., 2015*). Many circRNAs are differentially expressed in BC and participate in cancer-related pathways mainly by sequestering tumor suppressive miRNAs (*Lu et al., 2017*). Hippo signaling promotes BC progression by upregulating the expression of AMOTL1 and favoring metastasis (*Couderc et al., 2016*). Interestingly, the expression of circ-Amotl1 does not alter Amotl1 mRNA or protein levels. However, circ-Amotl1 interacts with c-myc and translocates to the nucleus, revealing that the functions of circ-Amotl1 are different from the conventional miRNA sponging activity of circRNA in BC cells (*Yang et al., 2017a*). Tumor metastasis is one of the most important factors for tumor death. Forkhead box C1 (FOXC1), the target of miR-3607, is downregulated in circIRAK3-silenced cells, and it mediates circIRAK3-induced BC cell migration (*Wu et al., 2018a*).

### Other diseases

Some other diseases also show a connection with circRNAs. Hsa_circ_0005105 promotes extracellular matrix degradation by regulating the expression of the miR-26a target, NAMPT, in osteoarthritis (OA) (*Wu et al., 2017*). Hsa_circRNA_103636 is easily detectable in blood samples, and the expression pattern of hsa_circRNA_103636 is altered in major depressive disorders (MDDs) (*Cui et al., 2016*). CircHECTD1 mediates silica-induced macrophage activation via HECTD1/ZC3H12A-dependent ubiquitination in Silicosis (*Zhou et al., 2018*). In bladder cancer, circTCF25 downregulates miR-103a-3p and miR-107 as well as upregulates cyclin-dependent kinases 6 (CDK6), suggesting that circTCF25 is a new biomarker (*Zhong, Lv & Chen, 2016*).

Cancer-related circRNAs with their corresponding miRNAs form a circRNA-miRNA-mRNA axis that regulates the expression of cancer-related proteins (*Sui et al., 2017*). Furthermore, the new expression of circRNAs in tumor cells, tissue-specificity, diversity, and high stability identify circRNAs as useful biological markers of cancer, thus improving the accuracy and specificity of diagnostic biomarkers (*Zhong, Lv & Chen, 2016*).

### CircRNA as a disease marker potential

Currently available reports clearly show that alterations in the expression of circRNA play important roles in the development of various pathological conditions. CircRNA is emerging as a novel biomarker due to its conservation, abundance, cell type-specific expression, tissue-specific expression, and roles in disease progression (*Meng et al., 2017*). HRCR attenuates the pathogenesis of cardiac diseases and has the potential to become a therapeutic target (*Wang et al., 2016*). CircFoxo3 may be a promising target of cellular senescence in aging heart (*Du et al., 2016a*).

Researchers have provided evidence that ciRS-7 has the potential as a biomarker for neurodegenerative disorder (*Lukiw, 2013*), diabetes (*Wang et al., 2013*), or MI (*Lin et al., 2018*). CircRNAs regulate the expression of cancer-related proteins in various types of cancer, including GC (*Li et al., 2015c*; *Shao et al., 2017*), CRC (*Wang et al., 2015a*),

ESCC (*Xia et al., 2016b*), HCC (*Yao et al., 2017*), and BC (*Lu et al., 2017*), through the circRNA-miRNA-mRNA axis, which represents their potential as diagnostic markers, prognostic indicators, therapeutic targets and even drug prospects in cancer.

## CONCLUSION AND FUTURE DIRECTIONS

CircRNAs have conserved sequences, tissue specificity, high stability and high abundance, thereby making them potential markers for disease screening and treatment (*Westholm et al., 2014*). The rapid development of high-throughput sequencing techniques and bioinformatics analyses suggest that circRNAs are likely to become new efficient targets in the clinical settings for the detection and treatment of diseases, such as diabetes, cancer, CVD, and neurological diseases. circRNAs can function as miRNA sponges (*Hansen et al., 2013*) and regulate multiple signaling pathways in CVDs (*Fan et al., 2017*), different types of cancers (*Lu et al., 2017*; *Meng et al., 2018*; *Yao et al., 2017*), neurodegenerative diseases (*Kumar et al., 2017*) and diabetes (*Zhao et al., 2017*). However, further studies are needed to reveal the complete biological functions of circRNA in terms of both physiological and pathological processes to promote the application of circRNA in future clinical use.

Despite rapid advances in the detection and characterization of circRNAs, the knowledge of circRNA functions is still at an early stage, which is one of the major drawbacks for the potential use of circRNAs for therapeutic or diagnostic purposes. New methods, such as chip technology, can be used to screen possible disease-related circRNAs in cell or experimental animal models, which will increase our knowledge about the role of circRNAs in the occurrence and development of pathological disorders. In addition to broadening functional aspects of circRNA, the following unknowns should also be addressed: mechanisms of trigger and control of circRNA formation dynamics; the link between the circRNA formation process and the corresponding linear RNA generation; the relationship between different circRNA products from the same gene; and regulatory mechanism of circRNA generation. The identification and characterization of specific circRNA-interacting molecules are important to provide information for most of these unknowns. In addition, the naming of circRNA has not yet been unified, and the mechanisms of circRNA in many diseases remain unclear. By solving these unknowns, circRNA may be a promising diagnostic tool and efficient therapeutic target for treatment of various pathological disorders.

### Funding

This work was supported by a grant from the National Natural Science Foundation of China Youth Fund Project (81522005); National Natural Science Foundation of China (81470522); National Natural Science Foundation of China (81500222); Shandong Province Natural Science Outstanding Youth Fund (2016JQB01015); China Postdoctoral Science Foundation (2017M612213). The funders had no role in study design, data collection and analysis, decision to publish, or preparation of the manuscript.

## Grant Disclosures

The following grant information was disclosed by the authors:
National Natural Science Foundation of China Youth Fund Project: 81522005.
National Natural Science Foundation of China: 81470522.
National Natural Science Foundation of China: 81500222.
Shandong Province Natural Science Outstanding Youth Fund: 2016JQB01015.
China Postdoctoral Science Foundation: 2017M612213.

## Competing Interests

The authors declare that they have no competing interests.

## Author Contributions

- Sheng Xu conceived and designed the experiments, performed the experiments, analyzed the data, contributed reagents/materials/analysis tools, prepared figures and/or tables, authored or reviewed drafts of the paper, approved the final draft.
- LuYu Zhou conceived and designed the experiments, approved the final draft.
- Murugavel Ponnusamy conceived and designed the experiments.
- LiXia Zhang analyzed the data, contributed reagents/materials/analysis tools, prepared figures and/or tables, authored or reviewed drafts of the paper.
- YanHan Dong performed the experiments.
- YanHui Zhang analyzed the data.
- Qi Wang contributed reagents/materials/analysis tools.
- Jing Liu authored or reviewed drafts of the paper.
- Kun Wang conceived and designed the experiments, approved the final draft.

## Data Availability

The raw data are included in the Figures.

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
