# Peer review of "A comprehensive review of circRNA: from purification and identification to disease marker potential"

_PeerJ, doi:10.7717/peerj.5503_

## Round 0.1 · original submission · Major Revisions

Please heed the comments of reviewers, especially reviewer #2, when amending your manuscript.

·

Basic reporting

The review gives a clear, well written account of the current understanding of CircRNA. While there are some minor errors, the authors give a very clear description of CircRNAs for people outside of the field, from their biogenesis, to their functions and future roles in molecular biology.

Experimental design

The review is structured into sections, first describing how CircRNAs are generated, followed by their function within the host system, followed by a list of methods of CircRNA detection including their costs and benefits depending on the method. Given what is known about this relatively novel RNA form, I feel the review gives appropriate time to each section, with appropriate citations. The sections are ordered appropriately for readers new to the field or with little knowledge of CircRNAs.

Validity of the findings

no comment.

Additional comments

The review is competently written and gives a good account of CircRNA for people new to the field. There are a few minor issues the authors could look to correct (I may have missed some).

47-62. The authors do not make clear how far reaching CircRNAs in their different forms are across the tree of life. Would it be possible to include a tree with presence/absence blocks for the differing forms of CircRNA (ecRNA/ciRNA/elciRNA) and their differing functions (miRNA sponge/RBP/pseudogene/regulation/splicing/translation), as I am unsure from the review is these forms/functions are ubiquitous or specific to some orders of life.

82-83. The sentence should be rewritten as 'Interestingly, not all exons form a circRNA...'

165: A small point, but it is a long time between RBPs being defined and first used, maybe restate here?

123-201: Another small point, but the order of sections here are confusingly ordered in relation to positions of illustrations in Figure 3. Possibly reorder the sections or labels within the figure.

Reviewer 2 ·

Basic reporting

The manuscript contains many typographic and drafting errors. English must be considerably improved. The manuscript includes sufficient background. It shows clearly the context.
The structure of the article is according to the required format by PeerJ. Figures are very illustrative, appropriately labelled and described.

Experimental design

The review is organized logically into coherent subsections. However, the “CircRNA in disease development and progression” section is poorly developed. The only paragraph in this section is difficult to understand and contains poor information.

Validity of the findings

No comment.

Additional comments

The review entitled “CircRNA: as a disease marker potential and research strategy” by Xu et al. addresses a very interesting and novel topic. In this review, authors explain relevant topics about CircRNAs including biogenesis, properties and functions. In general, these topics were correctly written and illustrated with appropriately described images. The “CircRNA in disease development and progression” section is not according to previous sections, this one is very poor. I consider that, since title is about CircRNAs and diseases, this section must be more extensive and complete. The “Methods of CircRNA detection and characterization” section is basically focused on methods of purification and identification of CircRNAs. This one is a small part that I consider to be a research strategy (as included in the title). Authors must discuss more about these topics and include Why CircRNA could be a disease marker potential?

I think that, if the authors address the previous comments and the following minor comments, the manuscript could be considered for publication.

Minor comment:

Section 7 is loss.

Line 48-49. Change “functional polypeptides to function” to “functional polypeptides”.

Line 88. Change “cricRNA biogenesis” to “circRNA biogenesis”.

Line 121-122. “CircRNAs play diverse biological roles due to the fact that average half-life of circRNA in most species is much longer than its linear counterpart.” What is the relationship between biological roles and longer half-life?

Line 127 . Change “cricRNA” to “circRNA”.

Line 130. Change “The systematically validated circRNA such as circRNA such as ciRS-7” to “The systematically validated circRNA such as ciRS-7”.

Line 141. Change “In generally, there are only few circRNAs” to “In general, there are only few circRNAs”

Line 157. Change “small nucleonucleo proteins (snRNPs)” to “small nuclear ribonucleo proteins (snRNPs)”

Line 180. Change “In eukaryotic system, the some circRNAs” to “In eukaryotic system, some circRNAs”
Line 218. Change “In particularly, their migration” to “Particularly, their migration”

Line 256. “cyclRNAs”. It is not the same nomenclature used through the manuscript.
Line 258. Change “The squence” to “sequence”
Line 268. Change “cricRNA” to “circRNA”.

---

## Round 0.2 · Major Revisions

The manuscript has beenimproved from the previous version, yet some problems remain that would benefit from a careful reread and, importantly, a very thorough checking of the English and grammar.

·

Basic reporting

In the presented manuscript, Xu et al present a review of what is currently known about circular RNA (circRNA), including their biogenesis, function and future directions for their use. While the review is of great interest, the authors have addressed many of the previous commends and the review provides a lot of information concerning circRNA, there is still room for improvement concerning the manuscripts writing.

Primarily, the authors make many claims throughout the manuscript not backed up by citations, maybe a further reread is required to insert correct or missing citations?

Additionally, as commented on below, the many of the sections are confusingly structured and the sentences within are poorly written. I feel the entire manuscript could benefit from a very careful reread and rewrite to clarify what is meant by each sentence, and to make sure the English is a lot clearer than it currently is.

Experimental design

no comment

Validity of the findings

As a review paper, the authors do not make any novel findings, however many of the claims of what circRNA can do are not backed up with citations, it would be useful if these citations could be provided.

Additional comments

I also have some comments on specific sections of the manuscript, I may not cover all grammatical errors and problems due to the sheer number of them.

Line 37: This sentence should be rewritten as: “Depending on their sequence, circRNAs can perform differing biological functions, including …. derivation of pseudogenes.”

Line 43: ‘Here we describe…’

Line 46: Is circRNA the correct pluralization? It reads strangely next to ncRNAs.

Line 55: this sentence should be rewritten as ‘Recent research shows that the lengths of mature circRNAs dictate…’

Line 70: Should this subheadings be here?

Line 72-109: The authors mention Alu elements several times during this section, they appear to be highly important for the generation of circRNAs but it is not clearly explained how important they are, maybe Alu elements could get a better description during this section?

Line 77-78: This sentence is very difficult to parse and should be rewritten entirely.

Line 86-88: This sentence is very difficult to parse and should be rewritten entirely.

Line 151-162: Maybe a better explanation of what species circRNAs are found in, and what species the examples given are in? As Ago2 is specifically siRNA in many organisms, and miRNA in only some.

Line 163-186: Should the ITCH system explained later also be mentioned in this section as a miRNA sponge?

Past this point there are many minor grammatical issues and confusing sentences, but no large problems, I recommend the authors still give the manuscript a thorough read through to fix the English as errors are present in every section throughout the manuscript.

Reviewer 2 ·

Basic reporting

No comment

Experimental design

No comment

Validity of the findings

No comment

Additional comments

The authors have adequately addressed most of the reviewers comments. However, the manuscript still contains many typographic and drafting errors (for example, line 256 “circ-Foxp3” instead of “circ-Foxo3” or line 308 “ripamycin” instead of “rapamycin”). English must be considerably improved.

Minor comments:

The sentences in line 286-288 and line 377-378 are not clear.

I don´t understand the sentence in line 334-335.

---

## Round 0.3 · Major Revisions

Dear Sirs:

I have read the newly submitted manuscript and unfortunately, in its present form, there are still many of the spelling, grammar, and English language defects that were noted in the previous two rounds of reviews. As this is the case, I am forced to ask for a very thorough review of the manuscript to assure that the spelling, grammar, and English are of a suitable quality for a scientific communication. Importantly, the overall clarity of the manuscript should also be addressed to improve readability, especially since this is a literature review article, geared for a more general readership.

If the command of English by the authors is not up to par, may I suggest contacting professional help to go over the manuscript and correct the many problems still there. Special care should be taken of number agreement between subjects and objects, for example, one of the many problems encountered. All of this has to be corrected before the manuscript is acceptable for re-review.

---

## Round 0.4 · accepted · Accept

Your review article has now been accepted for publication.

# ·

Basic reporting

The article reports on the function and recent findings concerning circular RNAs. The authors also report on circRNAs involvement in different diseases and cancers in humans. The article is written sufficiently to explain these factors, in an interesting fashion.

Experimental design

NA, the manuscript is as previous.

Validity of the findings

NA, the findings are as reported previously.

Additional comments

The english has improved drastically from the previous iteration of the manuscript, there are a few minor errors not worth reporting, following a reread by the authors, the manuscript is sufficient.

Reviewer 2 ·

Basic reporting

No comment

Experimental design

No comment

Validity of the findings

No comment

Additional comments

English has been considerably improved. The authors have adequately addressed most of the reviewers comments.